# Fast but not furious: a streamlined selection method for genome-edited cells

Haribaskar Ramachandran[1], Soraia Martins[1], Zacharias Kontarakis[2,3], Jean Krutmann[1,4,5], Andrea Rossi[1]

In the last decade, transcription activator-like effector nucleases and CRISPR-based genome engineering have revolutionized our approach to biology. Because of their high efficiency and ease of use, the development of custom knock-out and knock-in animal or cell models is now within reach for almost every laboratory. Nonetheless, the generation of genetically modified cells often requires a selection step, usually achieved by antibiotics or fluorescent markers. The choice of the selection marker is based on the available laboratory resources, such as cell types, and parameters such as time and cost should also be taken into consideration. Here, we present a new and fast strategy called magnetic-activated genome-edited cell sorting, to select genetically modified cells based on the ability to magnetically sort surface antigens (i.e., tCD19) present in Cas9-positive cells. By using magnetic-activated genome-edited cell sorting, we successfully generated and isolated genetically modified human-induced pluripotent stem cells, primary human fibroblasts, SH-SY5Y neuroblast-like cells, HaCaT and HEK 293T cells. Our strategy expands the genome editing toolbox by offering a fast, cheap, and an easy to use alternative to the available selection methods.

## Introduction

Genome editing technologies have substantially improved the ability to make precise changes in the genomes of eukaryotic cells. Programmable nucleases, such as meganucleases derived from microbial mobile genetic elements, zinc finger (Urnov et al, 2005; Miller et al, 2007), and transcription activator-like effector nucleases (Boch et al, 2009; Moscou & Bogdanove, 2009; Christian et al, 2010; Miller et al, 2011), have been used with discrete success to modify the genome of different species. More recently, the CRISPR/Cas9 from the type II bacterial CRISPR associated adaptive immune system revolutionized our ability to interrogate the function of the genome. It was shown to be potentially useful clinically in correcting genetic DNA mutations and to treat diseases that are refractory to traditional therapies or where therapies are not available yet (Pires et al, 2016; Ma et al, 2017; Min et al, 2019).

The overall success of the CRISPR/Cas9 system compared to the other genome editing technologies lies in its overall efficiency, low cost, straightforward plasmid assembly and an unmatched number of available DNA targets (Cong et al, 2013). The main limiting factor for many labs, when using the CRISPR/Cas system, is the ability to sort transfected cells that carry the desired mutation.

Nowadays, the enrichment of genetically modified cells after transfection is mainly achieved by antibiotic or fluorescent sorting. Both selection strategies are routinely used because they are relatively easy to perform and offer reproducible results. There are, however, several limitations that must be taken into consideration when choosing these selection methods. The most important ones are time, costs, and sorting efficacy.

To simplify the selection process, we thought to exploit the use of a magnetic-activated cell sorting system (Duda et al, 2014; Martin-Fernandez et al, 2020) to sort CRISPR genetically modified cells. Here, we describe a new method called "magnetic-activated genome-edited cells sorting" assay or MAGECS.

We show that MAGECS is a fast, easy, and relatively inexpensive pipeline to select genetically modified cells, which can be used to enrich CAS9-positive cells in different cell types including human-induced pluripotent stem cells (hiPSCs), primary human fibroblast, HaCaT, neuroblast-like cells and HEK 293T cells.

## Results

### Selection of genome-engineered cells

To select genome-engineered cells using MAGECS, we replaced the GFP sequence with a truncated CD19 domain and fused it downstream of the 2A peptide signal of the pSpCas9(BB)-2A-GFP plasmid (Fig S1). Transfected cells expressed a functional Cas9, able to translocate into the nucleus (Fig 1A) to mediate genome editing in the presence of a gRNA, as well as the surface marker tCD19, that enables sorting of transfected cells. We tested MAGECS by

[1]IUF-Leibniz Research Institute for Environmental Medicine, Core Unit Model Development, Düsseldorf, Germany   [2]Genome Engineering and Measurement Laboratory, Eidgenössische Technische Hochschule (ETH) Zurich, Zurich, Switzerland   [3]Functional Genomics Center Zurich of ETH Zurich/University of Zurich, Zurich, Switzerland   [4]Medical Faculty, Heinrich-Heine University, Düsseldorf, Germany   [5]Human Phenome Institute, Fudan University, Shanghai, China

Correspondence: Andrea.Rossi@IUF-Duesseldorf.de; zacharias.kontarakis@fgcz.ethz.ch

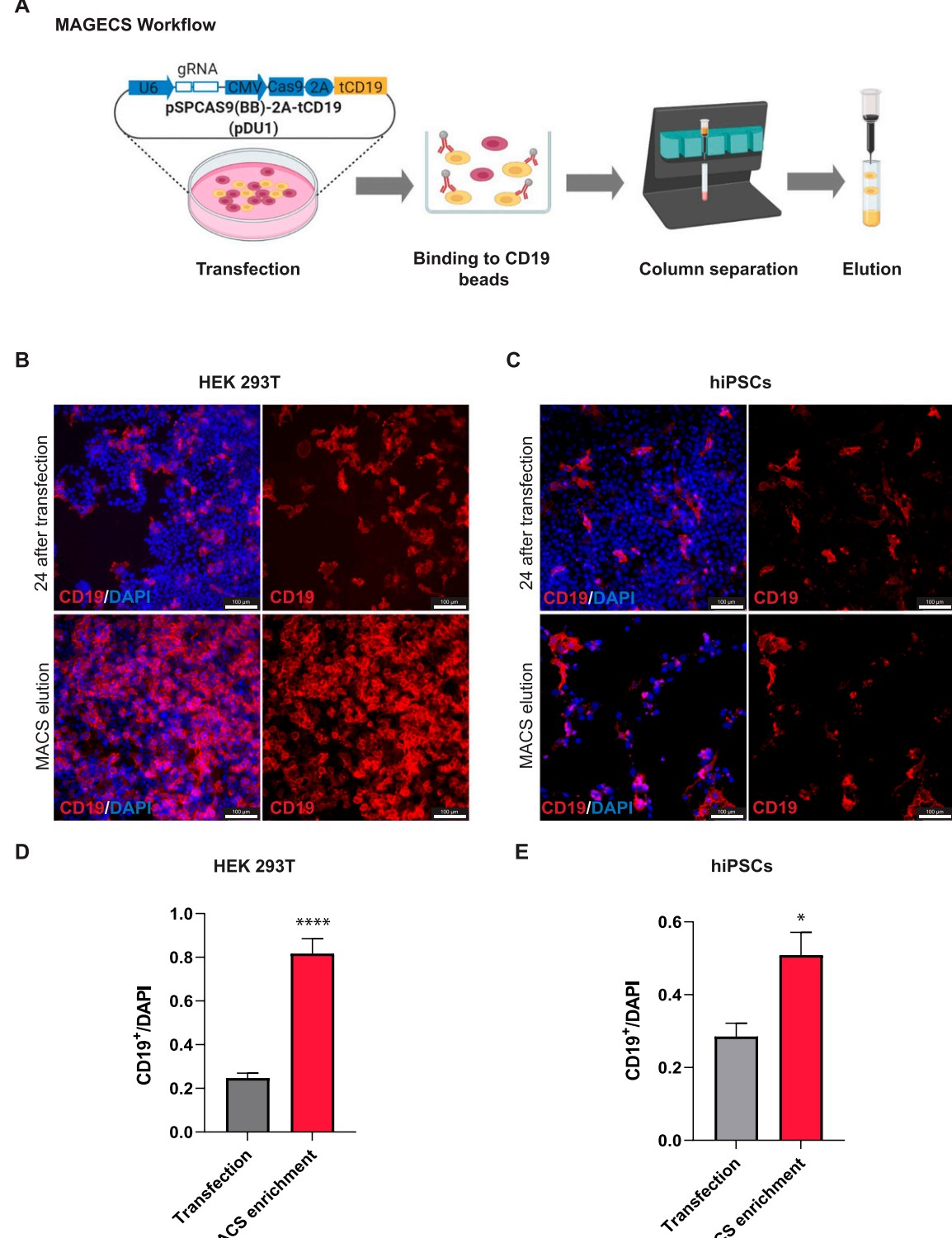

**Figure 1. tCD19 cell surface localization after transfection ensures efficient sorting.**
**(A)** Magnetic-activated genome-edited cell sorting workflow from transfection to elution. **(B, C)** Immunofluorescence of transfected HEK 293T cells (B) and human-induced pluripotent stem cells (C) using CD19 antibody showing the expression and proper localization of tCD19 motif. Cell elute with CD19 signal indicates the specificity of the whole process. **(D, E)** Quantification of the number of CD19 $^+$ cells after MACS. Results are mean + SEM from three technical replicates. Significance among conditions was calculated with two-tailed unpaired $t$ test. *$P < 0.05$ and ****$P < 0.0001$.

transfecting HEK 293T (Fig 1B) and hiPSC cells (Fig 1C) with the pSpCas9(BB)-2A-CD19 (pDU1) plasmid. Staining of transfected cells with CD19 antibody revealed the localization of CD19 to the plasma membrane. After magnetic sorting, the majority of eluted cells were CD19-positive cells (Fig 1B and C) indicating the efficiency and purity of CD19 selection, with an average of two to four times enrichment, when compared with unsorted cells, depending on the cell type (Fig 1D and E).

### Genome editing efficacy testing

Next, we designed and cloned different gRNAs targeting four human genes, *UDP Glucuronosyltransferase Family 1 Member A1* (*UGT1A1*), *dystrophin* (*DMD*), *actin beta* (*ACTB*), and *centrosomal P4.1-associated protein* (*CPAP*) into the MAGECS plasmid vector (pDU1). The MAGECS plasmids containing these gRNAs, were then transfected into HEK 293T cells and the cells were MACS sorted after 48 h (Fig S2A).

To assess the genome editing efficiency, we took advantage of the high-resolution melting curve analysis (or HRMA), which is routinely used because of its sensitivity, to detect known polymorphisms and particularly suitable for detecting indels induced by the genome editing technologies such as CRISPR/Cas9 system (Rossi et al, 2015).

After analysing wild-type (WT) and CD19-positive cells with HRMA a clear single peak was detected for WT and flow through cells indicating the lack of indels (Fig S3A and B). In contrast, CD19-positive cells displayed the typical irregular curve indicating the presence of DNA base indels (Fig S3A and B). Next generation sequencing and Sanger sequencing of CD19-positive cells confirmed the successful generation of indels indicating the efficacy of each gRNA and MAGECS to enrich genome-edited cells (Fig S2B and C).

### Cell viability after MAGECS

We next asked whether MAGECS would also be suitable for cell types beyond HEK 293T cells, that is, in particular primary cells, that are known to be more resilient to transfection and more sensitive to other sorting methods such as FACS.

To this end, primary human skin fibroblasts and SH-SY5Y neuroblast-like cells were transfected with gRNAs targeting *DMD* and *NADH:Ubiquinone Oxidoreductase Core Subunit S1* (*NDUFS1*), respectively (Fig 2A and D). Cells were magnetically sorted 2 d after transfection and divided in two halves. One half was plated on a new dish (Fig 2B and E) and the other one was used to extract genomic DNA for HRMA (Fig 2C and E). Indels were detected in CD19-positive cells by Next generation sequencing (Fig 2C and F). Furthermore, MAGECS did not reduce neither the viability of primary human fibroblast nor neuroblast-like cells as shown in Fig 2B and E.

**A**

| Gene | gRNA target | **PAM** |
|------|-------------|---------|
| *DMD* ex55 | GGTAGCATCCTGTAGGACAT | **TGG** |

**B**

**DMD mutant primary fibroblasts**

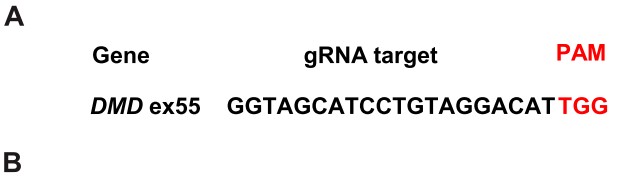

549μm

**C**

## *DMD* indels

WT  ACAACTGCCAATGTCCTACAGGATGCTACCCGTAAGGAAA

Δ5  ACAACTGCCAATGTCCT–––––ATGCTACCCGTAAGGAAA

Δ6  ACAACTGCCAATGTC––––––ATGCTACCCGTAAGGAAA

**D**

| Gene | gRNA target | **PAM** |
|------|-------------|---------|
| *NDUSF1* ex8 | CATCCACGGGATTGGAGATTC | **GGG** |

**E**

**NDUFS1 mutant Neuroblast-like cells**

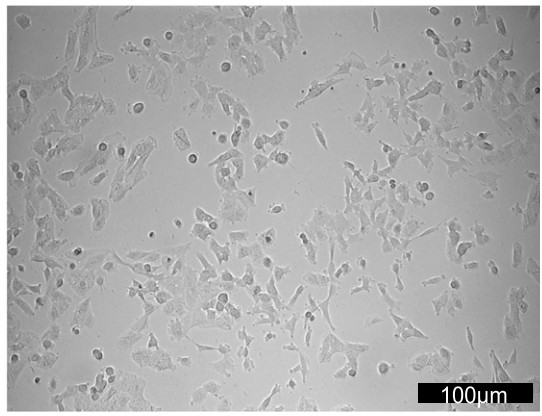

100μm

**F**

## *NDUFS1* indels

WT  ATATCATTGATATCTGCCCTGTAGGTGCCCTAACCTCTAAGCC

Δ1  ATATCATTGATATCTGCCCTGTAG –TGCCCTAACCTCTAAGCC

Δ1  ATATCATTGATATCTGCCCTGT –GGTGCCCTAACCTCTAAGCC

**Figure 2. Efficient sorting of genome-edited primary by magnetic-activated genome-edited cell sorting (MAGECS).**
**(A)** gRNA targeting human *DMD*. **(B)** Primary fibroblasts growing normally after MAGECS. **(C)** Sanger sequencing of clones confirming the deletion within *DMD*. **(D)** gRNA targeting human *NDUFS1*. **(E)** Viability of neuroblast-like cells is not affected by MAGECS. **(F)** Sanger sequencing of clones confirming the deletion within *NDUFS1*.

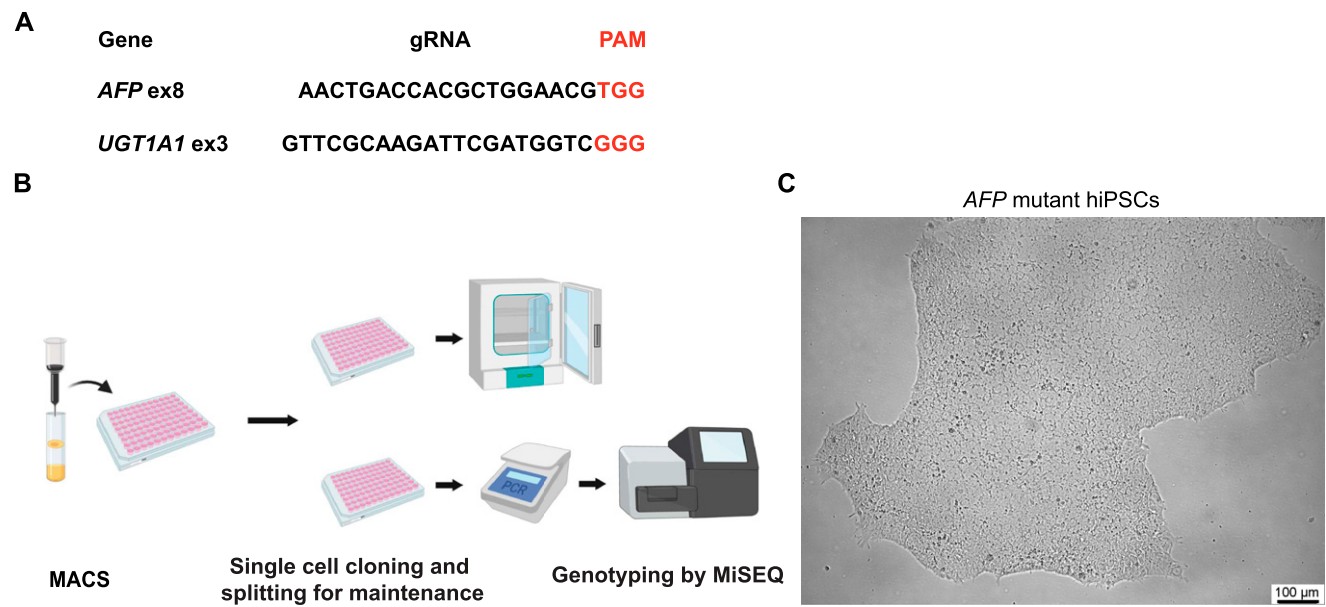

**A**

| Gene | gRNA | PAM |
|------|------|-----|
| *AFP* ex8 | AACTGACCACGCTGGAACG | TGG |
| *UGT1A1* ex3 | GTTCGCAAGATTCGATGGTC | GGG |

**B**

MACS — Single cell cloning and splitting for maintenance — Genotyping by MiSEQ

**C**

*AFP* mutant hiPSCs

100 μm

**D** *UGT1A1* indels

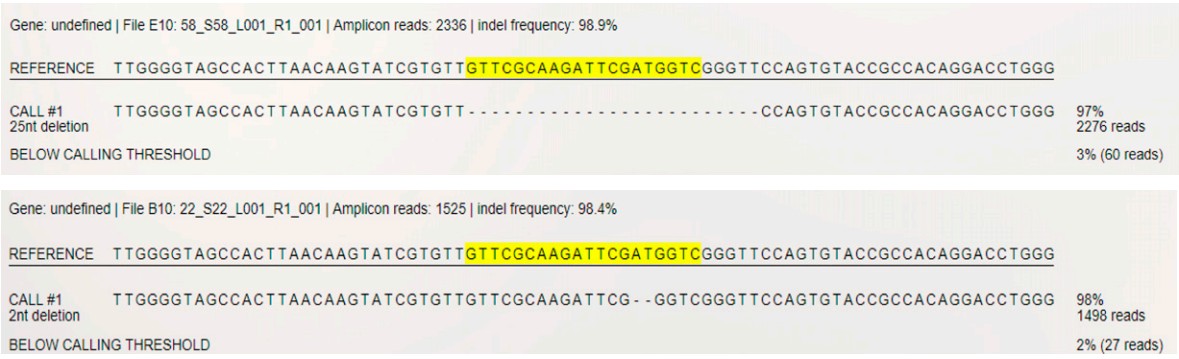

*AFP* indels

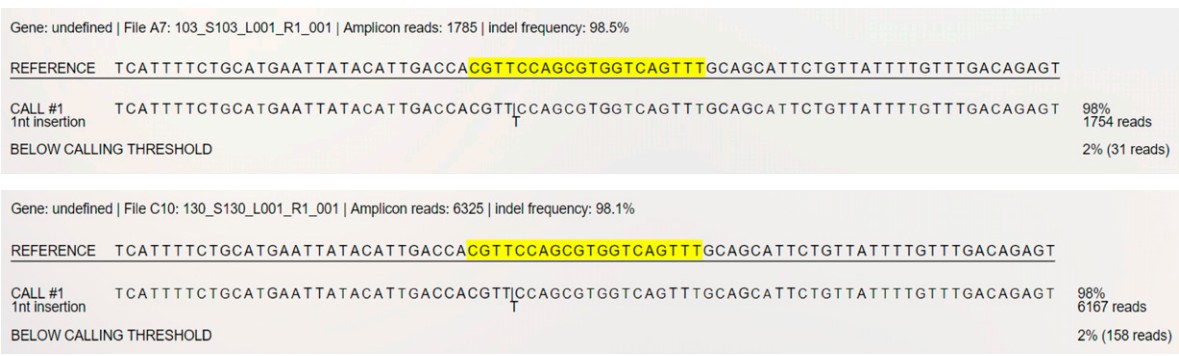

**Figure 3.   Efficient sorting of genome-edited human-induced pluripotent stem cells (hiPSCs) by magnetic-activated genome-edited cell sorting (MAGECS).**
**(A)** gRNA for human *UGT1A1* and *AFP*. **(B)** MAGECS protocol for hiPSCs. **(C)** Viability and morphology of *AFP* deleted hiPSCs is not affected by MAGECS. **(D)** MiSEQ analysis of positive clones for *UGT1A1* and *AFP* indicating the indel formation.

## MAGECS for sorting genome-edited hiPSCs

hiPSCs are difficult to maintain in culture and also to genetically modify their genome. Thus, we next asked whether MAGECS could be used to generate genome-edited hiPSCs of single clonal origin with a defined mutation. Human iPSCs were transfected with two gRNAs targeting *UGT1A1* and *alpha fetoprotein* (*AFP*) (Fig 3A) and magnetically sorted after 48 h. Single cells were plated in the presence of Rho Kinase (ROCK) inhibitors (i.e., Y-27632) onto a 96 multi-well plate and duplicated after 2 wk (Fig 3B). One plate was digested with Proteinase-K for further genotyping analysis and its duplicate was kept in the incubator. Deep sequencing analysis using an Illumina MiSeq sequencer (Schmid-Burgk et al, 2014) detected a high prevalence of mutations and revealed the presence of genetically modified cells carrying different types of mutations (Fig 3D), indicating that MAGECS can be used to successfully sort and genome engineer hiPSCs. The morphology of of iPSCs was not affected by MAGECS (Fig 3C). Furthermore, pluripotency and karyotyping analysis of hiPSC clones indicated that MAGECS protocol did not result in any chromosomal abnormalities and loss of pluripotency (data not shown).

# Discussion

A major challenge when engineering mutant models is the selection of genome-edited cells or single clones carrying the desired mutation. By combining two already available tools, the CRISPR/Cas9 system (Mali et al, 2013; Doudna & Charpentier, 2014; Cong & Zhang, 2015) and magnetic sorting (Miltenyi, 1990), we established a streamlined selection method of genome-edited cells called MAGECS. Using MAGECS, we were able to enrich genome-edited cells and select single clones for different cell types, including HEK 293T, HaCaT, human primary fibroblasts, and hiPSCs with different gene targets. Colony formation was observed after single-cell sorting by limited dilution, in all cell types tested, indicating that MAGECS does not affect the viability of cells. It also does not require expensive equipment or specialized personnel. The absence of genome-edited cells in the flow through further supports efficiency and accuracy of this method in capturing tCD19-positive cells (Fig S3).

Our assay offers several advantages compared with the available selection methods. In comparison to antibiotics MAGECS is faster and more consistent. Antibiotics are affordable and can be used without specific equipment, but the cell selection requires time and it is laborious when compared with other selection methods (i.e., FACS or MACS). The first critical step, when using antibiotics, is to determine the optimal concentration to use for the selection of stable colonies. The optimal concentration can be determined after generation of a killing curve, so that un-transfected cells are efficiently cleared without affecting the transfected ones. Furthermore, antibiotics require a few days to clear cells that are not carrying the antibiotic resistant gene, making the process of generating genetically modified cells longer. Last but not least, one of the main issues connected to the use of antibiotics is the possibility of spontaneously resistant clones that do not carry the gene of interest (Hawkey, 1998).

Antibiotics-free selection using FACS can be an alternative to avoid the issues mentioned above. Accordingly, the use of FACS was shown to significantly increase the number of cells that can be screened in a short period of time (Li et al, 2018). Unfortunately, the analysis of fluorescent markers requires large, complex, expensive instrumentation typically operated by highly trained specialists (Ren et al, 2019). The ability to quickly and simultaneously query multiple FACS parameters on large numbers of individual cells is generally reserved for shared core facilities or well-equipped specialized laboratories.

Occasionally, FACS-sorted cells fail to form colonies because of the exposure to a strong laser beam and the high hydrostatic pressure (Smith et al, 2006). When compared with FACS, MAGECS has the great advantage to be very convenient from an economical point of view. We have estimated that the cost of a single MAGECS-sorted sample is circa 12€. To sort cells using MAGECS, a strong magnet (i.e., OctoMACS Separator), MACS columns, and MACS microbeads are needed. The use of MAGECS allows us to sort up to eight samples in parallel, making the whole procedure very fast and medium throughput. In this regard, another advantage offered by MAGECS is that it can be performed in the same facility where the cells are stored by virtually any member of the scientific staff because it requires only minimal training. Thus, not only reducing the costs, but also the stress inflicted on the cells.

FACS and MACS are both very robust methods and they are able to produce results that overall are very consistent (Sutermaster & Darling, 2019). Nevertheless, the ability of MAGECS to isolate an increased cell number with high specificity in less time makes it an invaluable tool that will be useful for many scientists and labs working with genome editing.

In summary, we have demonstrated that MAGECS can sort genetically engineered cells with high efficiency and overall viability, and that it can be applied with minor protocol adjustments to a broad range of different cell types, including hiPSCs and primary cells. We expect that the lower cell loss associated with MAGECS will represent an advantage which might make MAGECS particularly useful for those "stress-sensitive" hiPSCs reprogrammed from patients carrying certain diseases.

We believe that MAGECS might have the potential to become a new standard for the streamlined selection for the genome editing of cells.

# Materials and Methods

## Cell lines

### Maintenance and passaging

Human fibroblasts, HEK 293T, and HaCaT cells were maintained in DMEM (Gibco) supplemented with 10% (vol/vol) heat-inactivated FBS (Gibco) and 1% (vol/vol) penicillin and streptomycin (P/S) (ref. P06-07100; Pan Biotech). SH-SY5Y neuroblast-like cells were maintained in DMEM: Nutrient Mixture F-12 (DMEM/F-12) (Gibco) supplemented with 10% (vol/vol) FBS and 1% (vol/vol) P/S. The cells were passaged using 0.05% trypsin (Gibco). hiPSC line IMR90 (WiCell) was cultured in mTeSR medium (Stem Cell Technologies) supplemented with 1% P/S on Matrigel-coated (Corning) plates. The medium was changed every day and the cells were passed every 5–6 d using ReLeSR (Stem Cell Technologies). 24 h prior transfection,

iPSCs were passed using Accutase (Innovative Cell Technologies) in a well of a Matrigel-coated six-well plate supplemented with 10 $\mu$M Y-27632 (Stem Cell Technologies).

## Plasmids

pSpCas9(BB)-2A-puromycin plasmid was purchased from AddGene (ref 62988). The plasmid that we generated, called pSpCas9(BB)-2A-tCD19 (pDU1) or MAGECS plasmid, was obtained by using the pSpCas9(BB)-2A-puromycin and swapping the puromycin with tCD19. Briefly, pSpCas9(BB)-2A-puromycin plasmid was digested with EcoRI and gel purified without the puromycin cassette. tCD19 was then amplified by PCR using primers with EcoRI adapters and cloned inside the gel purified pSPCas9(BB)-2A-puromycin back-bone. The plasmid was then fully sequenced.

## Transfection

Cells were transfected in six-well plates (TPP) with 2 $\mu$g of pSPCas9(BB)-2A-CD19 using FuGENE HD Transfection Reagent (Promega) or Lipofectamine Stem Transfection Reagent (Thermo Fisher Scientific), in the case of iPSCs, for 48 h according to the manufacturer's instructions.

## MAC sorting

Cells were sorted with MACS using MS columns, OctoMACS separator and CD19 MicroBeads (human) (all from Miltenyi Biotec) modifying slightly the data sheet protocol according to the experimental needs and the cell types (Fig 1).

After co-transfection, cells were detached (either with 0.05% trypsin or Accutase according to the cell type) and spun down at 500$g$ for 5 min (HaCaT and HEK 293T cells) or 300$g$ for 10 min (fibroblasts and hiPSCs). The pellet was then resuspended in 80 $\mu$l MACS buffer (PBS, pH 7.2, 0.5% FBS, and 2 mM EDTA) and 20 $\mu$l CD19 Miltenyi MicroBeads, mixed well, and incubated at 4°C for 15 min. After the incubation, 1 ml of MACS buffer was added to the cells and again spun down at 300$g$ for 10 min. The supernatant was discarded and the pellet was resuspended in 500 $\mu$l MACS buffer. After column equilibration, the cell suspension was loaded; CD19 negative cells passed through the column, whereas CD19-positive cells were retained. The retained cells were subsequently eluted using the syringe plunger to flush them trough. Cell suspension was spun down at 500$g$ for 5 min (HaCaT and HEK 293T) or 300$g$ for 10 min (fibroblasts, SH-SY5Y and hiPSCs), re-plated into six well plates or $\mu$-Slide 4 Well chambers (ibidi) to access the sorting efficiency.

## Cell lysis

The medium was discarded and the cells were lysed with the following of lysis buffer: 0.2 mg/ml proteinase-K, 1 mM $CaCl_2$, 3 mM $MgCl_2$, 1 mM EDTA, 1% Triton X-100, and 10 mM Tris (pH 7.5). Afterwards, the reactions were first incubated for 10 min at 65°C and then 15 min at 95°C.

## Generation of CRISP/Cas9 mutants

gRNAs were designed using CRISPR design tool (https://www.crisprscan.org/) and cloned into the MAGECS plasmid. The resulting bicistronic plasmid encoded the gRNA, the Cas9 nuclease, and the surface tCD19 marker. gRNA activity and efficiency were assed suing High-Resolution Melt Analysis (Rossi et al, 2015) using a MyGo PRO real time PCR (IT-IS Life Science LTD).

## PCR bar coding

PCR was performed as previously described (Schmid-Burgk et al, 2014). Briefly, first-level PCR reactions were performed using 1 $\mu$l PCR-compatible lysate as a template for a 6.25 $\mu$l Q5 (NEB) PCR reaction according to the manufacturer's protocol (annealing temperature: 60°C; elongation time: 15 s, 19 cycles) (NEB). Of this reaction, 2 $\mu$l was transferred to a second-level PCR using the same cycling conditions and a combination of barcode primers that is unique for each clone to be analysed. For all primer sequences see Table S1 and Supplemental Data 1.

## Deep sequencing

PCR products were pooled and size-separated using a 1% agarose gel run at 145 V. After visualization with SYBR Safe (Thermo Fisher Scientific) under blue light, DNA bands of around 450 bp were cut out and purified using GeneJET gel extraction kit according to the manufacturer's protocol (Thermo Fisher Scientific). DNA was eluted in water and then precipitated by adding 0.1 volumes of 3 M NaOAc (pH 5.2) and 1.1 volumes of isopropanol. After centrifugation for 10 min at 4°C, the resulting pellets were washed once in 70% EtOH and then air-dried. Afterwards, a total of 35 $\mu$l water was added, nonsoluble fractions were spun down and removed, and the DNA concentration was quantified using a Qbit4 spectrophotometer system (Thermo Fisher Scientific). Libraries were quantified using VAHTS Library Quantification Kit (Vazyme) and deep sequencing was performed according to the manufacturer's protocol using the MiSeq (Illumina) benchtop sequencing system. Data were obtained in FASTQ format and analysed with Outknocker (Schmid-Burgk et al, 2014).

## Immunofluorescence

For immunofluorescence staining, cells were fixed in 4% PFA-PBS for 10 min. After washing 3× with PBS, cells were blocked with 3% BSA diluted in PBS for 1 h at RT and then incubated with anti-CD-19-PE antibody (1:50) in 3% BSA-PBS (Miltenyi Biotec) for 1 h at RT. The nuclear stain Hoechst 33258 (2 $\mu$g/ml; Sigma-Aldrich) in PBS was added for 10 min. Fluorescent images were obtained using a fluorescence microscope Leica DMi8 (Leica). To determine the number of CD19[+] cells relative to the total number of cells, the total integrated density of the CD19 antibody was calculated and divided by the integrated intensity of the nuclei staining using ImageJ.

## Statistical analysis

Statistical analysis was performed with GraphPad Prism Software version 8.02 (GraphPad software). Two-tailed unpair $t$ test was used for statistical significance analysis for comparisons of the mean among conditions. Statistical significance was assumed at *$P$ < 0.05 and ****$P$ < 0.0001. Error bars represent mean + SEM of the experimental repeats.

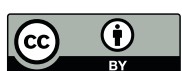
**Life Science Alliance**

# Supplementary Information

# Acknowledgements

We thank Carina Gude, Marie Brauers, Olivia van Ray, Björn Hiller, and Sara Desideri for technical assistance and help with the manuscript.

## Author Contributions

H Ramachandran: data curation, formal analysis, investigation, methodology, and writing—review and editing.
S Martins: data curation, formal analysis, investigation, methodology, and writing—review and editing.
Z Kontarakis: data curation, supervision, and writing—review and editing.
J Krutmann: funding acquisition and writing—review and editing.
A Rossi: conceptualization, resources, data curation, supervision, funding acquisition, investigation, methodology, project administration, and writing—original draft, review, and editing.

## Conflict of Interest Statement

The authors declare that they have no conflict of interest.

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
