## [Reviewer comments · Life Science Alliance]

Life Science Alliance

Fast but not furious: a streamlined selection method for genome edited cells

Haribaskar Ramachandran, Soraia Martins, Zacharias Kontarakis, Jean Krutmann, and Andrea Rossi

DOI: <https://doi.org/10.26508/lsa.202101051>

Corresponding author(s): Andrea Rossi, Dr. Andrea Rossi

Review Timeline:

Submission Date:	2021-02-09
Editorial Decision:	2021-03-29
Revision Received:	2021-04-01
Accepted:	2021-04-13

Scientific Editor: Shachi Bhatt

Transaction Report:

March 29, 2021

RE: Life Science Alliance Manuscript #LSA-2021-01051-T

Andrea Rossi

IUF-Leibniz Research Institute for Environmental Medicine, Genome Engineering and Model Development lab (GEMD)

Dear Dr. Rossi,

Thank you for submitting your manuscript entitled "Fast but not furious: a streamlined selection method for genome edited cells". We apologize for this extended and unusual delay in getting back to you.

As you will note from the reviewers' comments, both reviewers are quite enthusiastic about the MAGECS method, and have requested for only minor text and data presentations edits. We would be happy to publish your paper in Life Science Alliance pending final revisions necessary to address the minor concerns raised by both the reviewers and meet our formatting guidelines.

Along with the points listed below, please also attend to the following:

- please add ORCID ID for the corresponding author-you should have received instructions on how to do so
- please add your main, supplementary figure, and table legends to the main manuscript text after the reference section
- please rename the "Disclosure declaration" section to "Conflict of interest"
- please add the contribution of all authors to our system
- please add a Category and a Summary Blurb/Alternate Abstract for your manuscript in our system
- please upload your main and supplementary figures as single files
- please upload your Table in editable .doc or excel format
- please add callouts for Figures S2A, B, and S3A, B to your main manuscript text
- please improve the visibility of the scale bars in Figures 1B and C
- the NGS and Sanger sequencing used to confirm the generation of indels in the various cells is included in the Methods section or Data availability section. It is our policy to include accession numbers for large datasets generated in our published manuscripts. Please elaborate if there is any confusion or concern about sharing these.

A. FINAL FILES:

B. MANUSCRIPT ORGANIZATION AND FORMATTING:

Thank you for your attention to these final processing requirements. Please revise and format the manuscript and upload materials within 2 days.

Sincerely,

Shachi Bhatt, Ph.D.
Executive Editor
Life Science Alliance
<https://www.lsjournal.org/>
Tweet @SciBhatt @LSAJournal

Reviewer #1 (Comments to the Authors (Required)):

Haribaskar et al., described a new pipeline to sort genome edited cells that takes advantage of the ability to magnetically sort cells that express certain surface markers (eg CD19). In order to do so, the authors fused a tCD19 at the c-term of Cas9 (interspaced by a TA motif). At this point, they used magnetic sorting to sort genome edited cells from different cells types (including primary cells and iPSCs), highlighting the high efficiency and the easy to use of this method in sorting genome edited cells (SANGER and Deep Sequencing).

The manuscript is simple, well written and pretty straight forward in its intentions. I envision this method to be used in many labs, especially those with limited resources. I will also personally try this system out, considering how easier and straight forward it is compared to FACS sorting or antibiotics.

The figures seem a bit redundant, for instance Figure 2 could be used as a supplementary figure. Also, the manuscript mention the possibility to sort iPS cells deriving from human patients carrying certain diseases.

Did the author already try to perform such experiments? If so, the author should include these data in the manuscript.

Reviewer #2 (Comments to the Authors (Required)):

In this paper Ramachandran et al. established a selection method of genetically modified cells, called MAGECS, based on the ability to magnetically sort surface antigens in Cas9 positive cells. The protocol offers the advantage of being fast, cheap and easy to use and it can be applied to a variety of cells including primary cells and iPSCs. The experiments are clearly described and the manuscript is well written. I have only minor concerns:

- 1) The histogram in Supplementary Figure S2 should be included in Figure 1. It should be updated with statistical analysis and the statistics paragraph included in the Method section.
- 2) It is hard to assess that viability and morphologies of IPS are not affected by MAGECS by looking at Figure 4C. Please, substitute 4C image with two images (before and after MAGECS) at a higher magnification.
- 3) "MAGECS" should be included in the title.
- 4) Please revise all the manuscript for typos, in particular at the Method Section.

Reviewer 1

1. The figures seem a bit redundant, for instance Figure 2 could be used as a supplementary figure

We have revised all figures as suggested by both reviewers and moved Figure 2 from the main text to the Supplementary figures.

2. Also, the manuscript mentions the possibility to sort iPS cells deriving from human patients carrying certain diseases. Did the author already try to perform such experiments? If so, the author should include these data in the manuscript.

A very good idea indeed. We did not have the chance to use MAGECS for the sorting of genome edited patient-derived iPSCs. However, we have been successfully enriching patient-derived iPSCs using magnetic sorting. Thus, we envision MAGECS to be successful in generating genome edited patient-derived iPSCs.

Reviewer 2

1. The histogram in Supplementary Figure S2 should be included in Figure

1. It should be updated

with statistical analysis and the statistics paragraph included in the Method section.

We thank the reviewer for pointing out this issue, statistical analysis has now been added to the main Figure 1. We have also updated the Methods section and Figure legend 1.

2. It is hard to assess that viability and morphologies of IPS are not affected by MAGECS by looking at Figure 4C. Please, substitute 4C image with two images (before and after MAGECS) at a higher magnification.

We agree with the reviewer's comment, however to avoid having redundant pictures (see reviewer 1) in the manuscript, we have revised the main text and highlighted that both karyotyping and pluripotency analysis were performed and indicated that MAGECS does not affect chromosome stability neither pluripotency.

3. "MAGECS" should be included in the title.

We thank the reviewer for the suggestion, however we like the short and direct title and would like to keep it as it is.

4. Please revise all the manuscript for typos, in particular at the Method Section

We thank the reviewer for the pointing out this issue, we have now revised the whole Methods section.

Sincerely,

Andrea Rossi

April 13, 2021

RE: Life Science Alliance Manuscript #LSA-2021-01051-TR

Dr. Andrea Rossi
Dr. Andrea Rossi
Genome Engineering and Model Development
Auf'm Hennekamp 50
Düsseldorf
Düsseldorf, Nordrhein-Westfalen 40225

Dear Dr. Rossi,

Thank you for submitting your Research Article entitled "Fast but not furious: a streamlined selection method for genome edited cells". It is a pleasure to let you know that your manuscript is now accepted for publication in Life Science Alliance. Congratulations on this interesting work.

DISTRIBUTION OF MATERIALS:

Again, congratulations on a very nice paper. I hope you found the review process to be constructive and are pleased with how the manuscript was handled editorially. We look forward to future exciting submissions from your lab.

Sincerely,

Shachi Bhatt, Ph.D.

Executive Editor

Life Science Alliance

<http://www.lsjournal.org>
